# Tri-Training Algorithm for Adaptive Nearest Neighbor Density Editing and Cross Entropy Evaluation

**DOI:** 10.3390/e25030480

**Published:** 2023-03-09

**Authors:** Jia Zhao, Yuhang Luo, Renbin Xiao, Runxiu Wu, Tanghuai Fan

**Affiliations:** 1School of Information Engineering, Nanchang Institute of Technology, Nanchang 330099, China; 2School of Artificial Intelligence and Automation, Huazhong University of Science and Technology, Wuhan 430074, China

**Keywords:** Tri-training, training noise, nearest neighbor editing, local density, cross entropy

## Abstract

Tri-training expands the training set by adding pseudo-labels to unlabeled data, which effectively improves the generalization ability of the classifier, but it is easy to mislabel unlabeled data into training noise, which damages the learning efficiency of the classifier, and the explicit decision mechanism tends to make the training noise degrade the accuracy of the classification model in the prediction stage. This study proposes the Tri-training algorithm for adaptive nearest neighbor density editing and cross-entropy evaluation (TTADEC), which is used to reduce the training noise formed during the classifier iteration and to solve the problem of inaccurate prediction by explicit decision mechanism. First, the TTADEC algorithm uses the nearest neighbor editing to label high-confidence samples. Then, combined with the relative nearest neighbor to define the local density of samples to screen the pre-training samples, and then dynamically expand the training set by adaptive technique. Finally, the decision process uses cross-entropy to evaluate the completed base classifier of training and assign appropriate weights to it to construct a decision function. The effectiveness of the TTADEC algorithm is verified on the UCI dataset, and the experimental results show that compared with the standard Tri-training algorithm and its improvement algorithm, the TTADEC algorithm has better classification performance and can effectively deal with the semi-supervised classification problem where the training set is insufficient.

## 1. Introduction

Data mining refers to the process of finding specific rules and obtaining valuable information from data. Data mining methods include supervised learning, unsupervised learning, and semi-supervised learning [1,2,3]. Supervised learning uses labeled data to train classifiers and requires enough training data to ensure model performance; unsupervised learning does not require a priori information about the data and uses the idea of clustering samples with a high degree of similarity in the same clusters. Still, the accuracy of the model is difficult to guarantee. In the era of big data, the volume of data is growing exponentially, and it is easier to obtain large amounts of unlabeled data, which often requires a lot of human, material, and financial resources to assign labels to these data. Semi-supervised learning combines the advantages of supervised learning and unsupervised learning—breaking the boundary of traditional methods of using only one kind of data and using unlabeled data to assist labeled data for learning, which has become one of the research hotspots in the field of data mining [4]. Semi-supervised learning has been widely used in image processing [5], medical diagnosis [6], false comment detection [7], network security [8], etc.

Semi-supervised classification learning is a method that employs semi-supervised learning models to solve classification problems, which can guarantee the performance of classification models even when labeled data are insufficient. The main semi-supervised classification methods currently available include disagreement-based methods [9], generative methods [10], graph-based methods [11], and discriminative methods [12], etc. The disagreement-based method utilizes unlabeled data through multiple classifiers and adds pseudo-labels to the unlabeled data as a platform for information interaction among multiple classifiers during the training process. The generative method assumes that the data are all generated by the same generative model. The sample and class labels are generated by probability distributions of certain structural relationships, and the unlabeled data can be associated with learning objectives with the help of model parameters. A graph-based method maps data onto a connected graph by knotting geometric relationships between samples and constructing kernel functions to carry out information transfer of labels on the graph. The discriminative method learns decision boundaries by training both labeled and unlabeled samples through a maximum interval algorithm so that the learned classification hyperplane is at the maximum distance from the sample interval. Among them, the disagreement-based method is less affected by model assumptions and data size issues and is more widely applicable.

The semi-supervised classification methods based on the idea of disagreement originated from the co-training algorithm [13] proposed by Blum et al. The algorithm uses different views of labeled data to train two initial classifiers, and each classifier selects a number of unlabeled data with high confidence to add pseudo-labels to the training set of the other one and improves the classification performance by training each other. The algorithm requires at least two sufficiently redundant and conditionally independent views of the data, but in practical problems, it is difficult to satisfy this condition. To solve the problem of sufficiently redundant views of data, Zhou et al. [14] proposed the Tri-training algorithm, which generates three classifiers from a single-view training set to learn from each other to train the classification model. The algorithm does not require enough redundant views, which effectively reduces the data requirements and improves the efficiency and generalization ability of the algorithm.

Tri-training algorithm improves the performance of the classification model by expanding the training sets of three classifiers to each other; similar to the co-training algorithm, the performance of classifiers in the early iterative process is weak, and the multiple classifiers labeling the same unlabeled samples are prone to inconsistent class labeling. Secondly, the accumulation of mislabeled samples makes the performance of each classifier different, and the explicit decision mechanism using the voting method is prone to make the prediction results inaccurate.

To address the problem that expanding the training set is prone to forming training noise, Deng et al. [15] uses the remove-only editing operation to identify and remove the mislabeled samples that may arise from each iteration and proposed a Tri-training algorithm based on adaptive editing in combination with an adaptive strategy. Zhao et al. [16] use an active learning approach based on uncertainty and representativeness to sample unlabeled samples and proposed a group verification pseudo-label sample method and a security verification method., and select the labeled category with a low error rate as the final sample label by secondary validation. Li et al. [17] combine integrated learning with a Tri-training algorithm, estimate the confidence of unlabeled data by integrated learning, and use the confidence to weight the unlabeled samples to train the classifier, which reduces the influence of training noise on the classification model. Hu et al. [18] improved the access condition of the training set, estimated the prediction probability of unlabeled samples using the classifier, and dynamically adjusted the prediction probability threshold for the same unlabeled samples by tracking the sample distribution after each training round, reducing the noise while maintaining the sample class balance. Wang et al. [19] used an asymmetric tri-training model to expand the training set by estimating confidence and setting expert thresholds and combined the training sets of three classifiers to train a fourth classifier for decision target values using their common features and special features.

To address the prediction inaccuracy problem of explicit decision mechanism, Zhang et al. [20] introduced the ideas of cross entropy and convex optimization to improve the Tri-training algorithm, using cross entropy instead of the classification error rate to reflect the difference between the predicted distribution and the true distribution. The authors used the hinge loss function to estimate the degree of difference between the predicted and true values of the model and combined it with the convex optimization method to find the classifier weights under the minimum degree of difference to improve the accuracy of the model prediction. Mo et al. [21] assigned weights to the newly labeled data by Euclidean distance and entropy values, constructed a decision function based on the weight of each classifier’s accuracy and used a weighted voting method to predict the target values.

In order to better eliminate the training noise that impairs the learning ability of classifiers and improve the decision-making ability of classification models, this paper proposes a Tri-training for adaptive nearest neighbor density editing and cross-entropy evaluation (TTADEC). The structure of this paper is as follows: Section 2 outlines the Tri-training algorithm and its shortcomings; Section 3 discusses the new algorithm proposed in this paper in detail; Section 4 conducts simulation experiments on several UCI datasets and analyzes the performance of the algorithm; Section 5 concludes the paper.

## 2. Basic Concepts

### 2.1. Tri-Training Algorithm

Tri-training is a semi-supervised classification method based on the idea of divergence, which solves the problem of sufficient redundancy of data views in the co-training algorithm, which is not required for data views. The algorithm essentially uses a combination of multiple classifiers to combine the outputs of each classifier to obtain stronger performance than using a single classifier.

The basic idea of the Tri-training algorithm is as follows: first, three base classifiers are generated by randomly sampling training samples from single-view labeled data L through the bootstrap method; second, the training set of the three base classifiers is expanded, and any two base classifiers have the same prediction result for the unlabeled data sample x, then x is added to the training set of the third base classifier; finally, the base classifier is updated under the condition that the iterative process is satisfied until the base classifier error rate no longer changes and the training process is completed. The specific process is shown in Figure 1.

The Tri-training algorithm expands the training set to update the base classifier after each round of iteration to satisfy the constraints. Taking the expanded base classifier $h_{1}$ training set as an example, assume that $L^{t}$ and $L^{t-1}$ are the expanded training sets for rounds t and t−1, respectively, and the training noise rate $\eta^{t}$ of the base classifier $h_{1}$ in round t iteration is shown in Equation (1).
(1)$$\eta^{t}=\frac{\eta_{L}|L|+e_{1}^{t}|L^{t}|}{\left|L\cup L^{t}\right|}$$
where $\eta_{L}$ is the noise rate of the initial label data L and $e_{1}^{t}$ is the upper limit of the error rate of the joint classifier consisting of the base classifier $h_{2}$ and $h_{3}$ in the round t iteration.

According to the learnable theory in the literature [22], through Equation (2), the relationship between the training set size m, noise rate η and classification error rate ξ can be found as $\frac{1}{\xi }\propto m{\left(1-2\eta \right)}^{2}$.
(2)$$\frac{c}{\xi^{2}}=m{\left(1-2\eta \right)}^{2}$$

Zhou et al. [13] proved that the classifier performance is improved when the newly expanded training set has enough samples and the constraints of Equation (3) are satisfied.
(3)$$|L\cup L^{t}|{\left(1-2\frac{\eta_{L}|L|+e_{1}^{t}|L^{t}|}{\left|L\cup L^{t}\right|}\right)}^{2}>|L\cup L^{t-1}|{\left(1-2\frac{\eta_{L}|L|+e_{1}^{t-1}|L^{t-1}|}{\left|L\cup L^{t-1}\right|}\right)}^{2}$$
when $\left|L^{t}|>|L^{t-1}\right|$, Equation (4) can be introduced under the constraints of Equation (3):(4)$$0<\frac{e_{1}^{t}}{e_{1}^{t-1}}<\frac{\left|L^{t-1}\right|}{\left|L^{t}\right|}<1$$

The Tri-training algorithm expands the training set according to Equation (3), and the pseudo-labeled samples are labeled for base classifier $h_{2}$ and $h_{3}$ are added to the base classifier $h_{1}$ training set when the constraints are satisfied, and vice versa, the training set does not change. The training sets of base classifiers $h_{2}$ and $h_{3}$ are added in the same way until the iterative process does not satisfy the constraints, the training process ends, and the base classifier does not change anymore.

### 2.2. Nearest Neighbor and Relative Nearest Neighbor

Nearest neighbor is a classification algorithm based on the distance between samples in a neighborhood, which determines the class of samples to be tested by a voting method according to the nearest neighbor label.

**Definition** **1.***k Nearest Neighbor (KNN). Any sample point *i* in the dataset, the nearest *k* samples from which is the set of *k* nearest neighbors of sample *i*, denoted as *knn(i).

The relative nearest neighbor is then the correlation between sample k nearest neighbor. The shared area is part of the sample k nearest neighbor that is shared with the other sample k nearest neighbor, and the rest is the non-shared area, and the more samples in the shared area indicate the higher similarity between the two samples. The shared regions include shared nearest neighbors and natural nearest neighbors, which are defined as follows.

**Definition** **2.**
*Shared Nearest Neighbor (SNN). The set of *

k

* nearest neighbors of any sample *

i

* and *

j

* in the dataset, the set of nearest neighbors of sample *

i

* and sample *

j

* are *

knn(i)

* and *

knn(j)

* respectively, and the set of common nearest neighbors of sample *

i

* and sample *

j

* is called their shared nearest neighbor set, denoted as:*

(5)
SNN(i,j)=knn(i)∩knn(j)



**Definition** **3.**
*Natural Nearest Neighbor (NNN). Any sample *

i

* and *

j

* in the dataset, sample *

i

* and sample *

j

* have the set of *

k

* nearest neighbors as *

knn(i)

* and *

knn(j)

* respectively, *

knn(i)

* contains sample *

j

* and *

knn(j)

* contains sample *

i

*, then sample *

i

* and sample *

j

* are called natural nearest neighbor set and are denoted as:*

(6)
$$NNN(i,j)=\left\{\begin{array}{l} 1\;,\;i\in knn(j)\;and\;j\in knn(i)\;\\ 0\;,\;others \end{array}\right.$$



Shared nearest neighbors and natural nearest neighbors have been widely used in model optimization, machine learning and other fields. For example, Zhao et al. [23] defined the similarity between samples by shared nearest neighbors and natural nearest neighbors. The experiments showed that they could indicate the similarity degree of samples more accurately and improve the accuracy of sample assignment; Zhu et al. [24] incorporated shared nearest neighbors into single-cell clustering and defined the weights of edges by the ratio of shared nearest neighbors to nearest neighbors, and integrated the structural information of the graph to achieve the improvement of single-cell clustering accuracy. The samples within the shared region can be used indirectly as the similarity judging index. The more samples within the region indicate the higher similarity of the two samples and the higher possibility of belonging to the same kind.

## 3. TTADEC Algorithm

To address the shortcomings of Tri-training, the TTADEC algorithm is proposed in this paper. This section will introduce the TTADEC algorithm in detail, including the nearest neighbor editing, sample local density and adaptive sample expansion techniques used in the training process, the decision function of the algorithm and the algorithm steps.

### 3.1. Nearest Neighbor Editing

By analyzing the training process of the Tri-training algorithm, it can be found that the algorithm is prone to mislabeling unlabeled data as training noise in the early iteration process, which hinders the learning ability of the base classifier, resulting in the ineffective improvement of the base classifier’s performance. If the mislabeled samples can be identified and removed during the iterative process, more accurate hypotheses can be obtained, and the base classifier can predict the unlabeled samples more accurately afterwards.

For this purpose, the nearest neighbor idea is introduced to label the suspicious noisy samples in the iterative process. In most cases, the closer the samples are to each other, the higher their similarity and the higher the probability of belonging to the same class. The TTADEC algorithm expands the training set by considering the labeling of samples in the nearest neighbor range.

The specific idea is as follows: assume that the training set expanded from the unlabeled dataset U in the process of t iteration is $L^{t}$, and the initially labeled dataset is L. For each sample x in the training set $L^{t}$, select its k nearest neighbors knn(x) in $L\cup L^{t}$ according to the nearest neighbor rule, and observe whether the proportion of samples in knn(x) with the same label as x satisfies the threshold α. If the condition is satisfied, the accuracy of sample x with a pseudo-label is considered high, and the sample x in $L^{t}$ is labeled as a high-confidence sample; conversely, the accuracy of the sample adding a pseudo-label is considered low, and the sample x in $L^{t}$ is labeled as a suspicious noise sample.

### 3.2. Local Density of Relative Nearest Neighbors

The pre-selected samples of the expanded training set are obtained by sampling from the unlabeled dataset U. For sample regions that are in a dense degree, it is easier to use the nearest neighbor idea to determine whether the samples are suspicious noise, while the sparse sample regions do not work well using the nearest neighbor idea. As shown in Figure 2, let Figure 2a,b be the training data of rounds t and t+1, respectively, and the blue samples are pre-selected samples. When the pre-selected samples are in a dense area, it is more accurate to use the nearest neighbor approach to discriminate the noisy samples because the closer they are, the higher the probability of belonging to the same class of clusters. On the one hand, it is difficult to accurately predict the labels of the sparse samples in the initial iteration process because of the weak learning ability of the base classifier. This is due to insufficient training data; on the other hand, the distance between the sparse samples is large, the probability of belonging to the same cluster is low, and the nearest neighbor is easy to misidentify the pseudo labels. At the same time, as the number of iterations increases, the samples in the sparse region in the round t will change due to the continuous expansion of the training set. The denseness of the region they belong to will change afterwards, as shown in Figure 2c, the region where the yellow samples are located becomes more dense after the round t+1 iteration, and the inaccurate prediction of the unlabeled samples in the early iteration process will have an important impact on the labeling of suspicious noise samples afterwards.

Based on the above analysis, the TTADEC algorithm proposes the concept of sample local density, which is used to measure the density of the region to which the sample belongs, and does not label the expanded samples whose local density does not meet the threshold ρ, so as to reduce the problem of easy labeling errors in the early iteration process. The local density of samples needs to pay attention to the local information of samples, which can be reflected by the samples in the neighborhood. Therefore, the local density of samples can be determined by defining the similarity between samples and their neighbors, while the density difference between samples can be defined by the similarity between samples.

The sample similarity of the TTADEC algorithm considers the number of samples in the shared area and the distance between samples in the neighborhood range, while the sample local density is defined by summing up the sample similarity, and the formulas of sample similarity and local density are given below.

**Definition** **4.**
*Sample similarity. Any sample *

i

* and *

j

* in the dataset, the sample similarity is defined as follows:*

(7)
$$Sim(i,j)=\left(\frac{\left|SNN(i,j)\right|}{k}+NNN(i,j)\right)\left(\frac{\sum_{u\in [knn(j),j]}e^{-d_{iu}}+\sum_{u\in [knn(i),i]}e^{-d_{ju}}}{2(k+1)}\right)$$

*where *

$d_{ij}$

* is the Euclidean distance between samples *

i

* and *

j

*, the right bracket of *

Sim(i,j)

* reflects the density of the region in which the samples are located. The closer the distance, the larger the value and the denominator is the normalization parameter; *

|SNN(i,j)|

* the left bracket of *

Sim(i,j)

* is the number of shared nearest neighbors. *

NNN(i,j)

* is to determine whether samples *

i

* and *

j

* belong to natural nearest neighbors, shared nearest neighbors and natural nearest neighbors highlight the degree of similarity between samples, so the shared nearest neighbors and natural nearest neighbors are weighted to calculate the sample similarity, only when samples i and j have shared nearest neighbors or mutual natural nearest neighbors, there is similarity between samples.*


**Definition** **5.**
*Sample local density. The local density of any sample *

i

* in the dataset is the sum of the similarity of sample *

i

* to the rest of the samples.*

(8)
$$\rho_{i}=\sum Sim(i,j),j\in [1,2,\ldots ,N]$$

*where *

N

* is the number of pre-selected samples in the current iteration. A high local density of samples not only reduces the error rate when adding pseudo-labels to the base classifier, but also improves the accuracy of the nearest neighbor method in labeling suspicious noisy samples.*


### 3.3. Adaptive Sample Expansion

The TTADEC algorithm mechanically uses the nearest-neighbor idea and the local density of samples to expand the training set, which may sometimes damage the performance of the base classifier. If the sample size of a given iteration is not large enough, not only is the performance of the base classifier not improved, but the constraints of the iterative process are also not satisfied. In this regard, the TTADEC algorithm introduces an adaptive expansion of the training samples in terms of the mislabeling rate $a^{t}$.

**Definition** **6.***Mislabeling rate. The ratio of the number of incorrectly removed samples to the total number of removed samples for an arbitrary iterative process.*(9)$$a^{t}=\frac{\left|L^{t}|-|L_{n}^{t}\right|}{\left|L^{t}\right|}$$*where *$\left|L_{n}^{t}\right|$* is the number of samples for which the classifier predicts consistent results, and the degree to which the samples are screened indirectly affects the constraints under which Equation (3) holds. The number of noisy samples in the training set after the adaptive technique is *$e_{1}^{t}a^{t}|L^{t}|$*, which is brought into Equation (1) to obtain*.
(10)$$\eta_{de}^{t}=\frac{\eta_{L}|L|+e_{1}^{t}a^{t}|L^{t}|}{\left|L\cup L_{de}^{t}\right|}$$

**Theorem** **1.**
*In the round t iteration, *

$L_{de}^{t}$

* is the pre-selected sample set after nearest neighbor editing and local density screening, and the error rates *

$\xi^{t}$

* and *

ξdet

* after training the classifier using data *

L∪Lt

* and *

L∪Ldet

* satisfy the property *

$\xi_{de}^{t}<\xi^{t}$

* when the mislabeling rate *

$a^{t}\leq 1-\frac{\left|L^{t}|-|L_{de}^{t}\right|}{2e^{t}|L^{t}|}$

*. In the round t iteration, assuming that *

$L_{de}^{t}$

* is the pre-selected sample set after nearest neighbor editing and local density screening, when the mislabeling rate satisfies *

$a^{t}\leq 1-\frac{\left|L^{t}|-|L_{de}^{t}\right|}{2e^{t}|L^{t}|}$

*, the error rates *

$\xi^{t}$

* and *

ξdet

* after training the classifier using data *

L∪Lt

* and *

L∪Ldet

* satisfy the property *

$\xi_{de}^{t}<\xi^{t}$

* according to Equations (2) and (10).*


In summary, the classifier is trained on the filtered data samples when the mislabeling rate $a^{t}\leq 1-\frac{\left|L^{t}|-|L_{de}^{t}\right|}{2e^{t}|L^{t}|}$ during the iteration, otherwise, the classifier is not updated.

### 3.4. Decision Functions

The Tri-training algorithm uses “majority voting” to predict the target sample labels, and this explicit estimation makes the weights consistent across base classifiers. If the performance of the base classifier is poor, this explicit labeling confidence is not accurate enough, resulting in incorrect classification results. In most cases, the learning environment is not consistent across base classifiers, and it is difficult to obtain three classifiers with the same performance. The classifier with strong learning power cannot provide more information in the decision process, while the classifier with weak learning power tends to make the classification results worse. For this reason, For this purpose, the TTADEC algorithm decision process uses optimization methods to evaluate the performance of each classifier and assign appropriate weights to them.

The population intelligent algorithm is the mainstream method to solve optimization problems at present, which solves realistic optimization problems by simulating the group behavior of animals in nature, and the common population intelligence algorithms include firefly algorithm [25,26], particle swarm algorithm [27] and wolf pack algorithm [28,29], etc. Although population intelligence algorithms solve optimization problems with high accuracy, the corresponding time complexity is also high. Cross entropy is also a method for solving optimization problems, and compared with the population intelligence algorithm, the optimization method using cross-entropy is much lower in time complexity than the former. Most of the semi-supervised classification problems in real life are real-time problems, and for this reason, the TTADEC algorithm chooses cross-entropy to optimize the decision function.

The relative entropy [30] can be used to measure the variability of two probability distributions, and given that the true distribution of the sample set is P and the model predicted distribution is Q, the relative entropy of P and Q is defined as Equation (11).
(11)$$\begin{array}{ll} D_{KL}(P||Q)= & \sum_{i=1}^{n}P(x_{i})\log\left(\frac{P(x_{i})}{Q(x_{i})}\right)\\ & =\left(-\sum_{i=1}^{n}P(x_{i})\log Q(x_{i})\right)-\left(-\sum_{i=1}^{n}P(x_{i})\log P(x_{i})\right)\\ & =H(P,Q)-H(P) \end{array}$$
where H(P,Q) is the cross-entropy, and H(P) is the information entropy. The smaller the value of $D_{KL}$, the closer the distributions of P and Q. When $D_{KL}=0$, the distributions of P and Q are the same. In the actual classification problem, the true sample distribution P of the training data is known, the information entropy is kept constant, and only the cross-entropy H(P,Q) needs to be considered to evaluate the classifier performance.

In the TTADEC algorithm, the performance of base classifiers varies after learning, and the cross-entropy is used to evaluate the performance of each classifier and assign the corresponding weight to it, and the target sample is decided by weighted voting of classifiers. The cross-entropy of classifiers is calculated using the training set by extracting the dataset $L^{\prime }$ from the dataset L using bootstrap sampling, predicting the distribution of the dataset $L^{\prime }$ using the classifier $h_{i}$, and obtaining the cross-entropy $H_{i}$ from the true and predicted distributions of the dataset $L^{\prime }$. TTADEC, the weight function of the classifier, and the classification decision function in the algorithm are defined as follows:(12)$$\omega_{i}=1-H_{i}/\sum_{j=1}^{3}H_{j},i=(1,2,3)$$
(13)$$h(x)=\mathrm{sgn}\left(\sum_{i=1}^{3}\omega_{i}h_{i}(x)/\sum_{i=1}^{3}\omega_{i}\right)$$

Compared with the Tri-training algorithm, the TTADEC algorithm enables classifiers with strong learning power to provide more information through the classifier-weighted decision method and reduces the impact of weak classifiers prone to mislabeling, which improves the overall performance of the classification model to a certain extent.

### 3.5. Procedures

Algorithm 1 gives the pseudo-code of the proposed TTADEC algorithm., and the symbols are explained as shown in Table 1.
**Algorithm 1** Pseudo-code of the TTADEC**Input**: unlabeled dataset U, labeled dataset L, test set T.**Output**: h(x) (final training results).1: data preprocessing, data normalization and construction of distance matrix D.2: Li←bootstrap(L).4: **while** stopping condition is not met **do**5:         h(x)←update(hi).6:        selecting some samples U′ from the dataset U to add pseudo-labels.7:        obtaining the local density ρi of sample xi in U′ 8:        obtaining the k nearest neighbors of sample xi.9:        obtaining *L* through Equations (8) and (10).10:** end while**11: L′←bootstrap(L).12: H←∑h(L′)logy13: ω← based on Equation (12)14: h(x)← based on Equation (13)14: **return**
h(x).

## 4. Experiment and Analysis

### 4.1. Experiment Settings

To verify the effectiveness of the TTADEC algorithm, the algorithm performance was tested using the dataset in the UCI machine learning library [31]. We compare the TTADEC algorithm with the Tri-training algorithm [13], the Tri-training algorithm for adaptive data editing (ADE-Tri-training) [15], the Tri-training algorithm based on cross entropy (TCE) [20], the safe Tri-training algorithm (ST) [20], and the safe Tri-training algorithm based on cross entropy (STCE) [20] for comparison. The experimental environment is Intel(R) Core (TM) i5-6300HQ CPU @ 2.30 GHz, 12 G RAM, Windows 10 64-bit OS and Python 3.7 programming environment.

The experiments were conducted on nine datasets, and the basic information is shown in Table 2. These datasets are widely used to test various classification algorithms and vary in overall distribution, which can simulate different situations and compare the performance of algorithms in different scenarios. A total of 9 datasets have different sample sizes, number of attributes, class proportions, etc. Experiments on them can verify the generalizability of algorithms to different problems.

In this paper, four evaluation metrics are chosen to assess the algorithm performance, including accuracy, recall, precision, and F-measure. For semi-supervised learning, the data used in the experiments are few labeled data, and most of them are unlabeled data. Among them, 80% of the dataset is selected as the training set D and 20% as the test set T to verify the algorithm performance, 20% of D is the labeled dataset L, and 80% is the unlabeled dataset U.

The data were preprocessed before the experiment, including missing value processing and data normalization. Among them, missing values are filled by the average of valid data in the same dimension, and data normalization is performed by the maximum minimization method of Equation (14).
(14)$$x_{ij}{}^{*}=\frac{x_{ij}-\min(x_{j})}{\max(x_{j})-\min(x_{j})}$$
where $x_{ij}$ is the value of the j-th attribute in the i-th sample, and $x_{j}$ is the set of the j-th attribute in all the data. Data normalization is helpful to eliminate the influence of too large a gap between data of different dimensions on the experimental results.

### 4.2. Hyperparametric Analysis

In order to analyze the influence of the parameters α and ρ on the experiment, this paper uses the control variable method to select the appropriate parameters, in which the value of α is too small will make the confidence of the nearest neighbor editing too low, too high is easy to cause the number of pre-selected samples is not enough, for this control α value in 0.45~0.7 for the experiment; ρ value affects the probability of being mislabeled, control ρ value in 0~0.1 for the experiment.

Experiment 1 control ρ=0.06, the accuracy of the TTADEC algorithm was obtained on 9 datasets when α=0.45,0.50,0.55,0.60,0.65,0.70. The experimental results are shown in Figure 3. It can be seen from the Figure that, in most cases, the α value is more accurate in the interval 0.55~0.65. Experiment 2 control α=0.55 to obtain the accuracy of the TTADEC algorithm when ρ=0.02,0.04,0.06,0.08,0.10 on 9 datasets, and the experimental results are shown in Figure 4. It can be seen from the figure that, in most cases, ρ takes the highest accuracy when the value is in 0.06. In overview, considering the time cost of the algorithm is set to the median of the interval as 0.6 and is taken as 0.06 for subsequent experiments.

### 4.3. Performance Analysis

Experiments were conducted on nine UCI datasets, with α taking the value of 0.60 and ρ taking the value of 0.06, and the experimental results are shown in Table 3, Table 4, Table 5 and Table 6.

From Table 3, Table 4, Table 5 and Table 6, it can be concluded that on the three evaluation metrics of accuracy, recall, and F-measure, most of the datasets performed well on the TTADEC algorithm. On the accuracy metric, TTADEC has 8 datasets with the best results; on the F-measure metric, TTADEC has 7 datasets with the best results; on the recall metric, TTADEC has 6 datasets with the best results; and on the precision metric, TTADEC algorithm performs slightly worse with 5 datasets with the best performance.

To evaluate the comprehensive performance of each algorithm, the Friedman test [32] was introduced to evaluate the algorithms comprehensively. The larger the rank mean value of the Friedman test, the better the performance of the algorithm is indicated. Table 7 shows the rank mean values of each algorithm by the Friedman test.

From Table 7, it can be seen that the TTADEC algorithm has the highest rank mean value of all four evaluation metrics relative to other comparison algorithms, and the algorithm performance is TCE, Tri-training, ST, ADE-Tri-training, STCE, and TTADEC algorithms in order from low to high. Among them, the classification performance of Tri-training and TCE algorithms is poor; STCE and ADE-Tri-training algorithms have the middle performance; TTADEC algorithm has the most obvious classification effect.

Compared with the Tri-training algorithm, the ADE-Tri-training algorithm improves the training set expansion mechanism, and from the results of the rank mean, the adaptive editing expansion mechanism can improve the performance of the algorithm. TCE algorithm is to replace the joint classifier error rate during iteration, and from the results, the use of cross-entropy replacement error rate is less effective. ST algorithm constructs a function to optimize the decision weights in the algorithm’s decision process, and the effect of using this strategy alone is average, but it works better when used in combination with cross-entropy. TTADEC improves the ADE-Tri-training editing strategy by considering the sample information around the pre-selected samples and adding local density screening to expand the training set. This makes the added pseudo-labels less prone to errors, while the adaptive expansion will avoid the situation that the expanded training set is not large enough. The decision process to construct the decision function can also avoid the situation that poorly learned classifiers affect the prediction results, from the results show that TTADEC is the best performer.

In summary, combining the nearest neighbor editing technique of data density and the decision method of classifier weighting can improve the performance of classification models.

## 5. Conclusions

To address the poor performance of the early classifier of the Tri-training algorithm, the iterative process is prone to mislabeling the pre-selected samples in the expanded training set to form training noise and the inaccuracy of the explicit decision mechanism, this paper proposes the Tri-training algorithm for adaptive nearest neighbor density editing and cross entropy evaluation. The training phase of TTADEC algorithm screens pre-expanded samples by nearest neighbor editing and sample local density, dynamically expands the training set using adaptive techniques to reduce training noise during the iterative process, and the decision phase uses cross-entropy evaluation classifier to construct a decision function to improve the overall performance of the classification model. Experimental results on the UCI datasets show that the TTADEC algorithm has significantly improved the classification performance, effectively reduced the training noise formed during the classifier iterations, and improved the prediction inaccuracy of the explicit decision mechanism.

In this study, the training noise is reduced by the nearest neighbor idea in the training phase, which can show that supervised learning can assist semi-supervised learning for research; secondly, it is positive to study semi-supervised classification algorithms for industries such as e-commerce and network security, which can reduce data collection costs while ensuring accuracy.

In this paper, we use the nearest neighbor idea to improve the algorithm needs to rely more on the neighborhood information, and it does not perform well in classifying data with uneven density distribution. After that, we will improve the classification performance for data with poor density distribution, and also improve the correctness of suspicious noise selection. In the future, collective intelligence [33], swarm intelligence [34] or density peak clustering [35] approach may be used to settle the problem in our research.

## Figures and Tables

**Figure 1 entropy-25-00480-f001:**
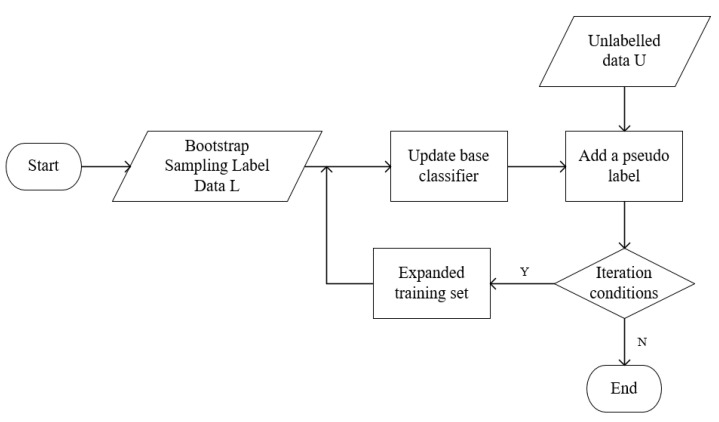
Flow chart of Tri-training algorithm training.

**Figure 2 entropy-25-00480-f002:**
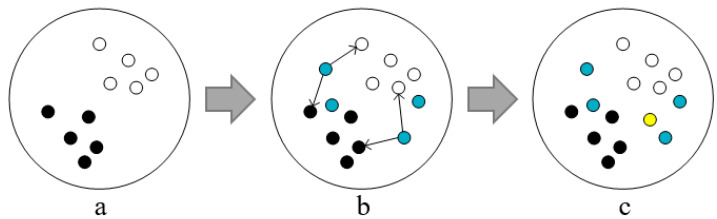
Schematic diagram of training set expansion. Where, figure (**a**) is the round t training set, figure (**b**) is the round t+1 training set, and figure (**c**) is the round t+2 training set.

**Figure 3 entropy-25-00480-f003:**
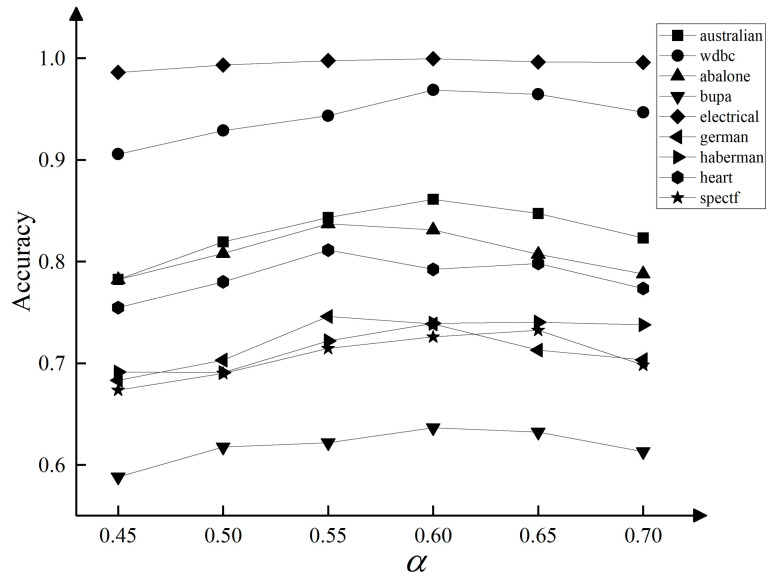
Effect of α on Accuracy.

**Figure 4 entropy-25-00480-f004:**
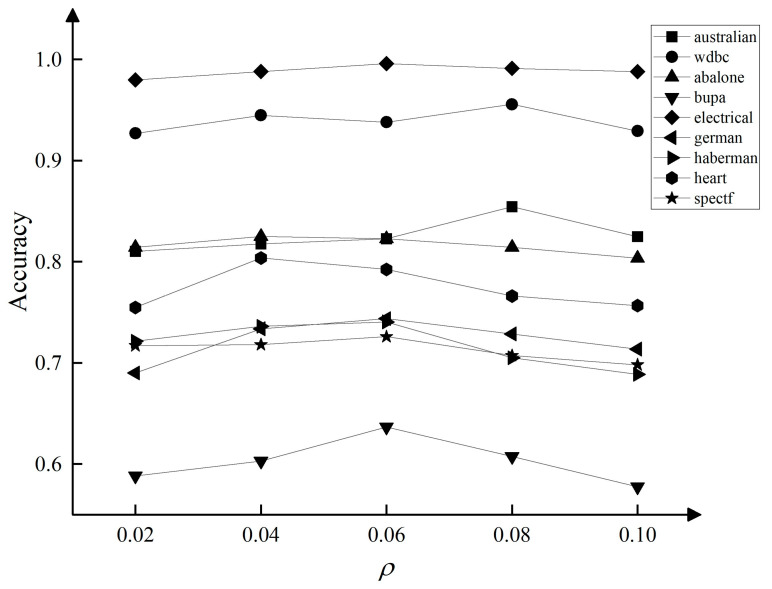
Effect of ρ on Accuracy.

**Table 1 entropy-25-00480-t001:** Symbol Description.

Symbols	Description
U	Unlabeled dataset
L	Labeled dataset
T	Test set
D	Distance matrix
$L_{i}$	Initial training set of each classifier
$\rho_{i}$	Local density
$a^{t}$	Mislabeling rate
$L^{\prime }$	Public dataset for weight acquisition
$\omega_{i}$	Weighting factor

**Table 2 entropy-25-00480-t002:** Experimental dataset.

Dataset	Size	Attribute	Positive	Negative
australian	690	14	44.5	55.5
wdbc	569	30	37.3	62.7
abalone	4177	8	32.1	67.9
bupa	345	6	42.0	58.0
electrical	10,000	13	36.2	63.8
german	1000	24	30.0	70.0
haberman	306	3	26.5	73.5
heart	270	13	44.4	55.6
spectf	267	44	20.6	79.4

**Table 3 entropy-25-00480-t003:** Accuracy.

Dataset	Tri-Training	ADE- Tri-Training	TCE	ST	STCE	TTADEC
australian	0.8022	0.8391	0.8206	0.8266	0.8497	0.8759
wdbc	0.9379	0.9363	0.9510	0.9441	0.9580	0.9735
abalone	0.8083	0.8198	0.7799	0.7837	0.8038	0.8371
bupa	0.5971	0.6097	0.5632	0.5517	0.5977	0.6364
electrical	0.9753	0.9900	0.9944	0.9940	0.9960	0.9995
german	0.7116	0.7227	0.7320	0.7400	0.7460	0.7437
haberman	0.7049	0.7044	0.5513	0.6923	0.5385	0.7405
heart	0.7632	0.7636	0.7206	0.7647	0.7794	0.8113
spectf	0.6415	0.7198	0.6418	0.5672	0.6269	0.7258

**Table 4 entropy-25-00480-t004:** Recall.

Dataset	Tri-Training	ADE- Tri-Training	TCE	ST	STCE	TTADEC
australian	0.7422	0.8083	0.7765	0.8077	0.7826	0.9038
wdbc	0.9359	0.9533	0.9857	0.9718	1.0000	0.9646
abalone	0.6120	0.6240	0.5455	0.5551	0.6143	0.6607
bupa	0.6126	0.6927	0.8000	0.6818	0.6875	0.7357
electrical	0.9746	0.9990	0.9945	0.9923	0.9935	0.9995
german	0.6750	0.4454	0.5067	0.5185	0.5652	0.7553
haberman	0.3693	0.2636	0.1724	0.2778	0.1875	0.4846
heart	0.7165	0.8236	0.6471	0.7097	0.7059	0.7912
spectf	0.6147	0.4102	0.3429	0.2778	0.3421	0.6356

**Table 5 entropy-25-00480-t005:** Precision.

Dataset	Tri-Training	ADE- Tri-Training	TCE	ST	STCE	TTADEC
australian	0.6685	0.8713	0.8462	0.8077	0.9231	0.7966
wdbc	0.8866	0.9457	0.9200	0.9200	0.9200	0.9777
abalone	0.6408	0.7128	0.6094	0.5898	0.5352	0.7114
bupa	0.3276	0.2880	0.2553	0.3191	0.4681	0.5166
electrical	0.9577	0.9989	0.9902	0.9913	0.9956	0.9996
german	0.8833	0.6462	0.5588	0.6176	0.5735	0.8659
haberman	0.2333	0.3053	0.3125	0.3125	0.3750	0.4333
heart	0.9112	0.8048	0.7586	0.7586	0.8276	0.9285
spectf	0.6928	0.8751	0.9231	0.7692	1.0000	0.9523

**Table 6 entropy-25-00480-t006:** F-measure.

Dataset	Tri-Training	ADE- Tri-Training	TCE	ST	STCE	TTADEC
australian	0.7490	0.8356	0.8098	0.8077	0.8471	0.8268
wdbc	0.9359	0.9488	0.9517	0.9452	0.9583	0.9689
abalone	0.5644	0.6206	0.5756	0.5720	0.5720	0.7258
bupa	0.5126	0.4634	0.3871	0.4348	0.5570	0.5660
electrical	0.9746	0.9920	0.9923	0.9918	0.9945	0.9993
german	0.5450	0.7835	0.5315	0.5638	0.5693	0.7577
haberman	0.2843	0.2320	0.2222	0.2941	0.2500	0.4571
heart	0.7410	0.7016	0.6984	0.7333	0.7619	0.8333
spectf	0.5458	0.6484	0.5000	0.4082	0.5098	0.7478

**Table 7 entropy-25-00480-t007:** Rank mean values of indexes in six algorithms.

Evaluation Indicators	Tri-Training	ADE- Tri-Training	TCE	ST	STCE	TTADEC
Accuracy	2.44	3.44	2.56	2.67	4.00	5.89
Recall	2.89	3.89	2.78	2.78	3.22	5.44
Precision	2.67	3.89	2.56	2.67	4.00	5.22
F-measure	2.44	3.78	2.33	2.50	4.28	5.67
Mean	2.61	3.75	2.56	2.65	3.87	5.55

## Data Availability

Not applicable.

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
