# Peer review of "Tri-Training Algorithm for Adaptive Nearest Neighbor Density Editing and Cross Entropy Evaluation"

_entropy, 2023, doi:10.3390/e25030480_

Round 1

Reviewer 1 Report

Tri-training algorithm for Adaptive Nearest Neighbor Density Editing and Cross Entropy evaluation

This paper proposes a Tri-training for adaptive nearest neighbor density editing and cross entropy

evaluation (TTADEC) to better eliminate the training noise that impairs the learning ability of classifiers and improve the decision-making ability of classification models.

Results: TTADEC algorithm has better classification performance compared with the standard Tri-training
algorithm and its improved algorithm

The paper is good in general, but I suggest that the author should add the figure of ROC (Receiver operating characteristic) in order to compare visually the performance of these classification algorithms. This kind of figure will help readers to easy to recognize which method is performed better than others.

Besides, there are many machine learning models perform very well for classification problems, the authors should discuss about this.

The authors should add the Discussion part.

Discussion: in the context of previous work (with references) discuss how your results corroborate, contradict or provide novel knowledge. Focus the discussion on responding to the original objectives and identify limitations of the work and future research.

- The discussion section needs to be described scientifically. Kindly frame
it along the following lines:
i. Main findings of the present study;
ii. Comparison with other studies;
iii. Implication and explanation of findings;
iv. Strengths and limitations.

In the Conclusions: please address in at least three separate and brief paragraphs the following: i) main findings of the paper; ii) limitations of this work and future research; iii) broader impacts (what others in the field or different fields can do with the findings presented in this work).

Reviewer 2 Report

In this paper, the authors proposed a tri-training algorithm for adaptive nearest neighbor density editing and cross-entropy evaluation (TTADEC). The proposed algorithm used the nearest neighbor editing to label the high-confidence samples in the training process. It defined the sample local density of the relative nearest neighbor to screen pre-training samples. Then, it dynamically expanded the training set by adaptive technique to reduce the training noise formed in the classifier iteration process. The decision process used cross entropy to evaluate the completed base classifier. Besides, it assigned appropriate weights to construct a decision function to reduce the influence of training noise in the classifier prediction stage. The authors should consider the following points in their revised paper:

1.       The paper needs intensive proofreading as it contains many long, inconsistent sentences that are hard to follow.

2.       The abstract section is inconsistent and does not reflect the main contributions of the manuscript. The authors should rewrite the abstract section to mention the paper's main purpose, primary contributions, experimental results, and global implications using short, concise sentences.

3.       On page 2 (line 54), What is the author mean by "inaccurate"?

4.       In the introduction section, the authors should explain the discussed studies in the simple past tense.

5.       The paper contains many punctuation errors, such as capitalizing many unnecessary words throughout the paper.

6.       Write the word "dataset" using only one form throughout the paper.

7.       The authors should rename the second section as "basic concepts" or "Tri-training algorithm," not "related work."

8.       It is highly recommended to list all used symbols in a table to track the proposed algorithm easily.

9.       On page 9 (line 319), what is value in "In this paper, four evaluation metrics are chosen to assess the algorithm performance, 318 including accuracy, recall, precision, and value."?

10.   The authors should write a new section that discusses their experimental results.

Round 2

Reviewer 2 Report

Thanks very much to the authors for their effort in improving their manuscript. They satisfied most of my comments. Besides, I do not have more comments for them.